# Climate Change and Natural Resource Scarcity: A Literature Review on Dry Farming

Naomi di Santo [1] , Ilaria Russo [2] and Roberta Sisto [1,*]

1 Department of Economics, Management and Territory, University of Foggia, 71121 Foggia, Italy
2 Department of Humanities Literature, Cultural Heritage, Education Sciences, University of Foggia, 71121 Foggia, Italy
* Correspondence: roberta.sisto@unifg.it

**Abstract:** The agricultural sector is facing the challenge of climate change, which is increasing difficulties to the activity and the economic sustainability of the primary sector, also affecting farmers' revenues. There is a growing need to support policy makers' decisions and help them develop cross-sectional strategies to support farmers. To this aim and to collect useful information for policy makers and stakeholders for the development of efficient strategies for the management of dryland farming, the paper examines how this issue has been analysed in the literature. A mixed method, based on a systematic literature review and a bibliometric analysis of 79 Scopus documents using VOSviewer software, was applied. Major results highlight the need to implement participatory policy interventions so as to include farmers. It was possible to summarise the main adaptive and technical interventions implemented by farmers. The results indicated the importance of the concept of the resilience of territories and the need to analyse agricultural systems by considering their multifunctionality. The innovativeness of this study relies on its relationships with several policy aspects and not only with purely technical and agronomical features, analysing thus the issue from the under-investigated perspective of the global challenge, contributing to filling this literature gap.

**Keywords:** climate change; natural resource scarcity; dry farming; dryland farming water scarcity

## 1. Introduction

Many authors have highlighted the key role agriculture plays in the sustainable development of a country [1,2] as well as for the development of rural areas, the preservation of biodiversity, economic growth, and food security [2–4].

In recent years, the agricultural sector has faced the challenge of climate change, which is causing an increase in the severity and frequency of extreme weather events and natural disasters [5,6]. Climate change and the agricultural sector have a two-way relationship. More specifically, the agricultural sector is compromised by the effects of climate change, such as the increase in global temperatures and changing precipitation levels and patterns [7]. On the other hand, agricultural activity negatively impacts climate change, e.g., by increasing greenhouse gas emissions [8]. This generates a vicious loop that increases the difficulties of activities in the primary sector and emphasises the need to find new ways to safeguard the entire sector [9]. Moreover, by affecting agricultural yields, climate change also affects farmers' revenues (increasing income volatility) and the economic sustainability of the whole sector. Therefore, it is important to structure and implement adequate and responsive strategies involving either risk management tools or agronomical practices [10]. Moreover, it becomes necessary to stimulate the application of innovative programmes optimising the use of scarce natural resources [11].

According to previous studies [12,13], the greatest impact of climate change will occur in global arid and semi-arid regions. It is crucial to consider this issue because arid and semi-arid ecosystems occupy more than 3 billion hectares and are home to 2.5 billion

people [14]. Consequently, the irrigated land available for agricultural production could decrease because of severe water scarcity and the limited availability of new land [13,15].

Because of climate change, natural resources are becoming increasingly scarce, as their rate of regeneration is higher than their rate of consumption. This is particularly noticeable for water. In fact, one of climate change's impacts is the reduction of the occurrence of precipitation, and when it occurs, it happens with very high intensity and time concentration [16]. To cope with this issue, agriculture requires the development of new techniques and management practices [17], such as so-called dryland farming or dry farming [18], a production without irrigation in dry seasons, especially in regions receiving a maximum of 50 cm of rainfall in a year [19]. Surprisingly, drylands are also common in developed countries, characterised by an increased risk of desertification [20]. For example, thirteen European countries have declared to be affected by desertification [20]. For this reason, several European institutions are dealing with desertification issues—e.g., the European Parliament and Council introduced some measures to promote the efficient use of water in agriculture in the Common Agricultural Policy (Re. UE 1305/2013; e.g., by recycling and re-using water resources) [21]. From the literature [22,23], it emerges that Southern Europe is at a higher level of desertification risk than the whole European area, and new strategies are being implemented to cope with water scarcity, such as the use of cover crops to help conserve water in the soil. Moreover, climate change is impacting the agricultural sector by changing cultivation patterns, which also influences international trade, exposing farmers to greater economic losses [24].

All this highlights the weaknesses of the agricultural sector, which is notoriously one of the riskiest production activities because its results are heavily affected by natural conditions such as precipitation. Nowadays, this riskiness is emphasised by the need for sustainable use of natural and scarce resources and has increased because of the growing deterioration of soil fertility, the loss of some important ecosystem services (including biodiversity and wild foods), and the threat of increased climate variability. For these reasons, there is a growing focus on strategies enabling farmers to adapt to current challenges [25,26], especially in arid and semi-arid areas [27]. However, the literature appears to be fragmented and focused only on certain geographic territories, not considering that an increasing area will become arid over time [28].

In recognition of this scenario, the aim of this study is to collect useful information for policy makers and stakeholders who have to develop efficient strategies for the management of dryland farming. The goal of this paper is achieved through the implementation of a systematic literature review on Scopus, one of the major search engines. Synthesising the aspects analysed in various fields allows a representation of reality, giving attention to the needs and risks of the sector to outline and structure specific new policies.

In particular, after an analysis of the current state of the literature from main bibliographic databases such as Scopus, a bibliometric review was implemented. In this case, VOSviewer software was chosen, so the research is based on an objective and replicable analysis. Researchers have already used this software extensively for the analysis of several complex problems; in fact, searching "VOSviewer" on Scopus gives more than 3000 results. On the other hand, the application of bibliometric reviews and the use of this software appear to be poorly implemented in this area of research. The strengths of the VOSviewer software are the easy management of a large number of articles and the ease of clustering the results derived from the search engines in small groups, allowing an intuitive and better synthesis of the results from the literature review.

To the best of our knowledge, various authors have investigated the mentioned issue in different fields of research [2,29,30], but studies focusing on policy implementation are lacking. Therefore, this paper is intended to fill the literature gap—most of the studies on this topic are almost technical. The innovation of this study is linked to advice for policymakers, including economic and financial aspects, about the main needs resulting from climate change and has already been highlighted by researchers of other disciplines. The structure of the paper is as follows: the methodology is described in Section 2, followed

by Section 3, in which the results are presented. Sections 4 and 5 are dedicated to the results' discussion and concluding remarks, respectively.

## 2. Materials and Methods

The study was carried out using a mixed approach based on a systematic literature review followed by bibliometric analysis. In particular, to develop a replicable and objective analysis of the study, a keyword search was deployed on Scopus, while the VOSviewer software was used to implement the bibliometric analysis. Hence, a query was defined with the operator "TITLE-ABS-KEY". The main keywords used in the search were "dryland farming" or "dry farming" and "climate change".

The Scopus database was chosen for several reasons, the main ones concerning the attention given to the peer-review procedure and the accessibility of key information and bibliographic data about publications [31]. In addition, compared to other search engines, it appears to have a larger time coverage that allows for the inclusion of citation analysis and study evolution [32,33]. Moreover, the ease of use is greater than other search engines. In fact, it is possible to export data directly into a form acceptable by many types of software dedicated to bibliometric analysis [31]. To examine how the researchers analysed the phenomenon over time, all available years were included. For this reason, no inclusion or exclusion criteria were applied regarding the period. Moreover, considering that the study is intended to guide policy choices regarding the management of this multidisciplinary topic, no subject areas were excluded. Finally, only open-access articles in the English language published in journals were considered, increasing the opportunity for replication of the study by all those interested in the topic. In conclusion, 79 papers were found to be suitable for the analysis. The paper selection process is shown in Figure 1.

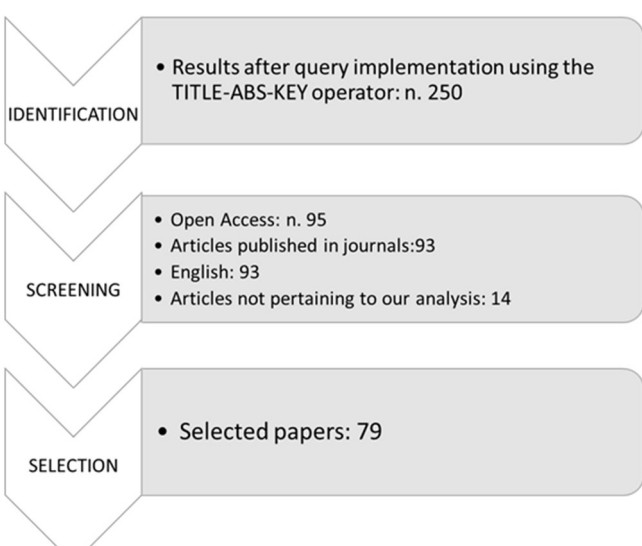

**Figure 1.** Flow chart of the articles' selection process. Source: our elaboration.

A bibliometric analysis was performed using VOSviewer software, a useful tool to map and graphically represent the relational structure of documents sourced from various search engines. The software identifies clusters of documents according to their correlation degree, so every paper is assigned to a specific cluster, allowing the identification of the main themes considered by the literature. Each cluster has a specific colour, giving a visual identification of relationships between papers. This allows an immediate representation of state of the art, even in the presence of a high number of publications.

## 3. Results

First, the distribution of scientific articles over time was analysed, and the results are shown in Figure 2.

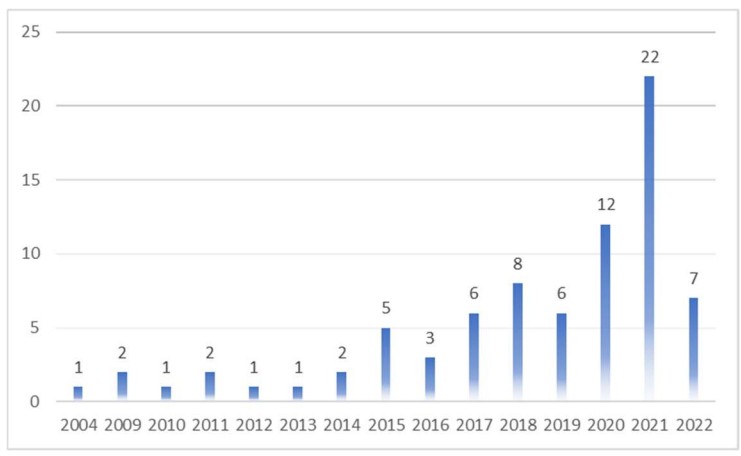

**Figure 2.** Distribution of the papers over time. Source: our elaboration.

Analysis results show that the selected papers were published between 2004 and 2022; the current year (2022) was included in this study because a large number of papers were congruent with this analysis. On the other hand, an important publication gap emerged from 2004 to 2009. Moreover, Figure 2 shows an increasing trend. Specifically, from 2004 to 2016, the considered topic was discussed in one or two papers annually, with a maximum of five papers published in 2015. However, since 2017, the minimum number of papers published annually has never fallen below six papers. These results highlight the relevance that this issue is receiving in literature, also driven by the many policy documents that put environmental issues at the top of their policy agenda.

Separately analysing research trends related to climate change and dryland farming on the Scopus search engine shows some interesting aspects. As far as climate change, before 2006, there was limited interest in this issue. However, interest increased in relation to an emerging and positive trend of research (Figure 3A). With respect to dryland farming, papers on this topic show an increasing trend, starting from 2015 (Figure 3B) (a possible explanation is the 2030 Agenda signed in 2015).

Then, the analysis was focused on the main journals in which at least two papers were published (Table 1), while the complete list of these journals is provided in Table A1 of the Appendix.

The journals that include the largest number of publications are Climate Change and Sustainability (Switzerland). The variety of topics covered by other journals pushes attention to the multidisciplinary nature of the topic. In fact, other issues largely addressed by those journals are, for example, sustainability, environmental management, risk management and engineering.

Moving on, the analysis was focused on the relationships between the most widely used keywords in the papers. This was conducted to better explore the links between the keywords, thus providing a greater understanding of the state of the art. The lowest allowable number of keyword occurrences was fixed at five, so groups were shaped by considering the keywords appearing together at least five times. Figure 4 shows the most commonly occurring keywords: "climate change", "dryland farming", and "climate effect" or "crop yield"; less common keywords include "irrigation", "desertification", "adaptive management", or "vulnerability".

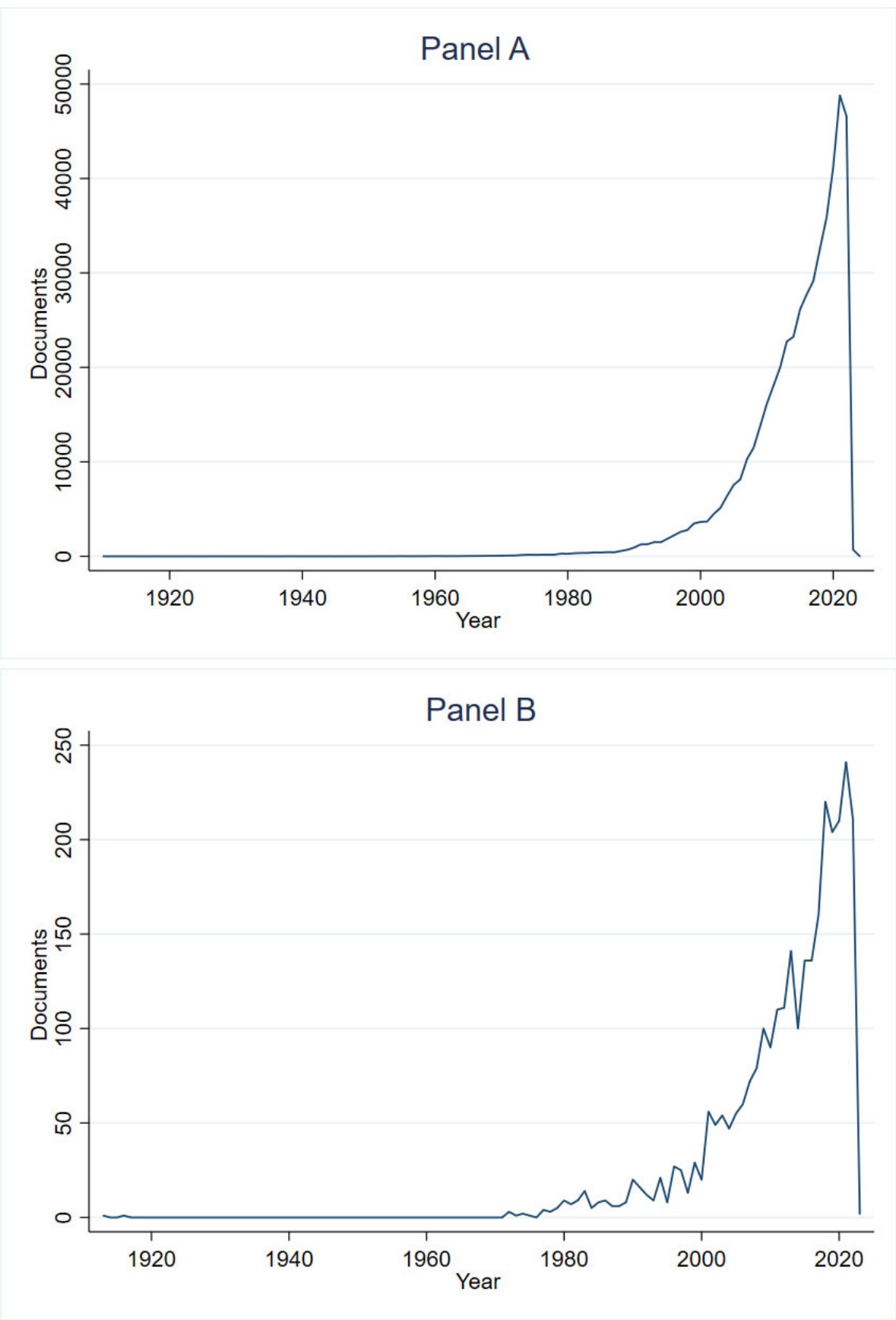

**Figure 3.** Trend in research about "climate change" (**A**) and "dry farming" or "dryland farming" (**B**). Source: Scopus results.

**Table 1.** Major journals publishing studies on this topic.

| Journal | Number of Papers |
| --- | --- |
| Climatic Change | 6 |
| Sustainability (Switzerland) | 6 |
| Nature Communications | 4 |
| Agricultural Systems | 3 |
| Ecology and Society | 3 |
| Environmental Research Letters | 3 |
| Land Use Policy | 3 |
| Agriculture, Ecosystems and Environment | 2 |
| Current Opinion in Environmental Sustainability | 2 |
| International Journal of Water Resources Development | 2 |
| Journal of Ecology | 2 |
| Land Degradation and Development | 2 |
| Science of the Total Environment | 2 |
| Water (Switzerland) | 2 |

Source: our elaboration.

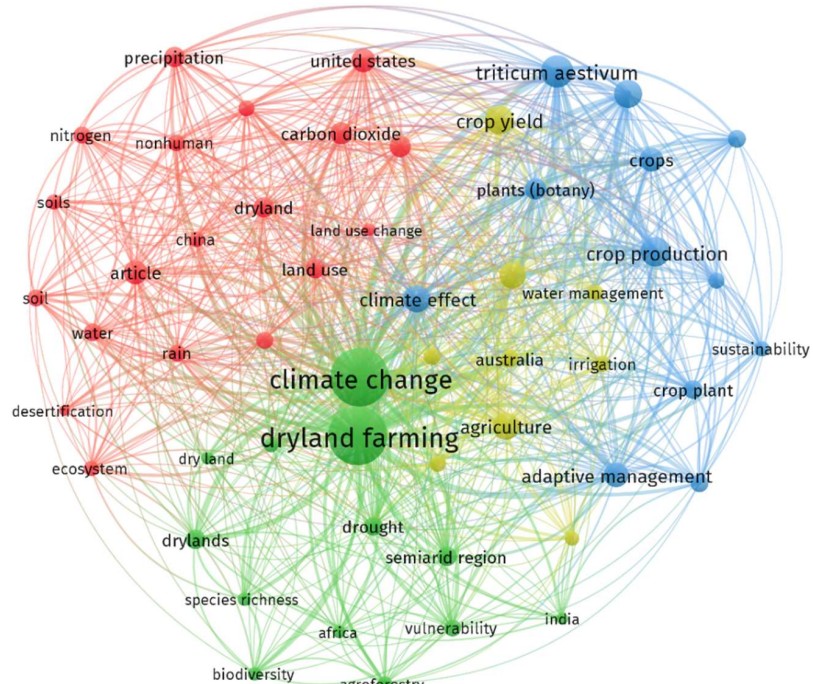

**Figure 4.** Keyword Network Mapping. Source: VOSviewer elaboration.

*3.1. VOSviewer Analysis*

The graphic result of the VOSviewer software is shown in Figure 5. More specifically, this analysis divided the search field into five clusters.

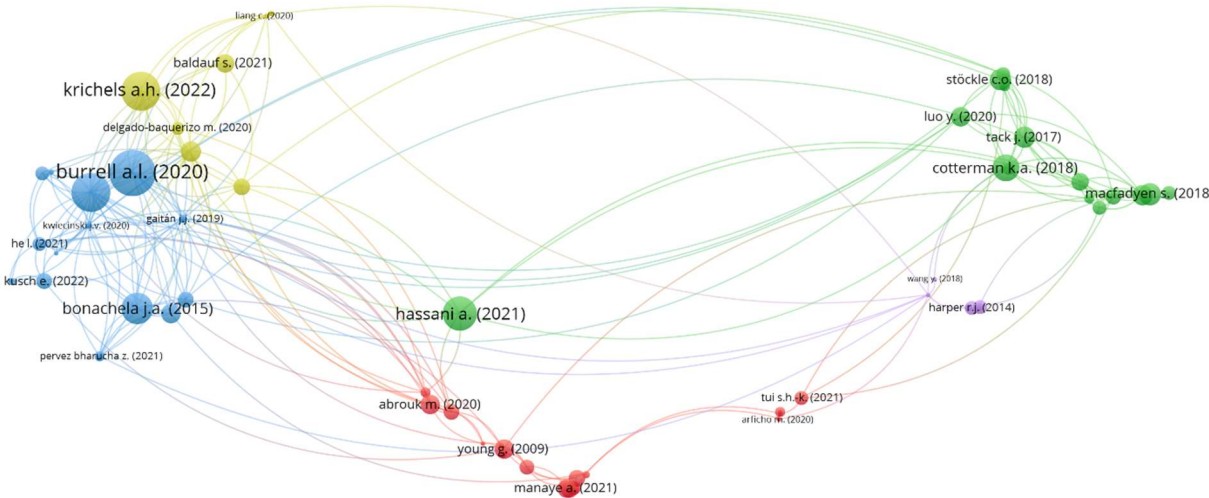

**Figure 5.** Results of the VOS analysis. Source: VOSviewer elaboration.

Figure 5 shows that the clusters are very distinct from each other, highlighting different views of the issue. Therefore, several aspects will be taken into consideration. Descriptive statistics for each cluster are reported in Table 2, while bibliographic data for each paper are given in the Appendix (Table A2).

**Table 2.** Cluster descriptive statistics.

|  | Number of Papers | Total Citations | Total Normalised Citations | Total Citations/ Number of Articles |
|---|---|---|---|---|
| Red Cluster | 21 | 361 | 99.804 | 17.19 |
| Green Cluster | 16 | 447 | 168.262 | 27.93 |
| Blue Cluster | 16 | 439 | 156.207 | 27.44 |
| Yellow Cluster | 9 | 88 | 41.134 | 9.77 |
| Purple Cluster | 6 | 178 | 3.321 | 26.67 |

The Red cluster includes the largest number of items (21), followed by the Green and Blue clusters, both of which include 16 documents. More specifically, the Green cluster has the highest number of total citations (No. 447), and it is the most relevant when considering the ratio of total citations over number of articles.

The different clusters can be summarised as follows:

- The Red cluster, titled "Farmer inclusion and policy interventions";
- The Green cluster, titled "Potential adaptations to climate change";
- The Blue cluster, titled "The resilience of arid areas";
- The Yellow cluster, titled "A variety of indices";
- The Purple cluster, titled "The multifunctionality of agricultural systems".

### 3.1.1. Red Cluster: Farmer Inclusion and Policy Interventions

The Red cluster includes the largest number of papers (21). Specifically, only one paper had not a specific area of study. The remaining twenty were focalised as follows: fourteen papers analysed data on the African continent, four papers were focused on the Asian continent, and the European and American continents were analysed by one paper. This divergence is due to the fact that climate change is one of the major threats affecting food production and the delivery of products and services, especially for developing countries [34].

Bibliometric analysis shows that the paper with the most normalised citations was written by Abrouk et al. [35], analysing the fonio genomic resources, which is the first step toward exploiting the potential of this cereal crop for agriculture in poor environments. Indeed, white fonio is a native West African millet species with characteristics that make it a strategic crop for agriculture in marginal environments [36,37]. Abrouk et al. [35] reveal the key characteristics that make this cereal suitable for arid and semi-arid soils, such as drought resistance and adaptation to nutrient-poor soils, including sandy soils [38].

The importance placed on projects and policy intervention emerges in this cluster [39–41]. Specifically, it is clear from various papers [16,25,42] that the inclusion of farmers is a necessary strategy to address new spatial and environmental challenges and to have timely and lasting results. For example, the 'Farmer-Managed Natural Regeneration' approach, which aims to restore and reforest arable land by working together with farmers, was analysed [43]. In this strategy, groups of the most dynamic farmers in each community have been judged essential for the mobilisation of other community members to learn and experiment [25]. The first result of this approach is a set of psychosocial benefits, such as joy and peacefulness, resulting from the increased beauty and comfort of the greener landscape; increased confidence and experience in leadership for the stakeholders that took part in the groups; improved attitudes toward environmental management and increased optimism for the future of their farms and communities [25].

The projects that have been taken up in these clusters, therefore, promote stakeholder inclusiveness in the area of interest, improved livelihoods, the development of resilient and productive ecosystems, and the development of governance systems that support farmers [39,44].

The literature [25,42,45,46] shows the importance of agroforestry, which is also emphasised in many National Adaptation Programmes of Action (NAPA). Reintroducing trees and shrubs has many benefits: (a) to improve soil nutrients [47]; (b) to increase water infiltration into the soil; (c) to reduce soil evapotranspiration and reduce soil temperature; (d) to increase wildlife diversity, including animal, bird and insect species [48]; (e) to protect the livelihoods of farming households [49]; (f) to diversify income to woody and non-woody tree products [50]; and (g) to improve household resilience to drought or pests [51].

Another aspect emerging from the literature review concerns the techniques that farmers implement to adapt to climate change [27]. Indeed, new water management techniques have been developed to restore soil moisture (referred to as green water), water collection through storage structures (referred to as blue water) [52], new anti-erosion techniques, and mulching [25].

In conclusion, dryland agriculture is a complex and vulnerable system composed of crops (cereals), vegetables, livestock, and trees. Therefore, managing risk and productivity through the adoption of resilient technologies and sustainable intensification is critical to securing income and improving livelihoods in these vulnerable regions [16].

3.1.2. Green Cluster: Potential Adaptations to Climate Change

The Green cluster includes 16 papers concerning a common topic: climate change adaptation strategies.

Indeed, with growing worries about climate change's future influence on agricultural production, many scholars [53–55] have used quantitative and qualitative methods to analyse the strategies farmers adopt to respond to new environmental challenges.

Table 3 summarises the interventions that emerged from the literature. It is important to specify that some of these strategies also emerged in the Red cluster but less frequently. In fact, Figure 5 shows a significant relationship between the two clusters, anticipating the overlap of some themes.

**Table 3.** The adaptation strategies.

| The Adaptation Strategies | | |
|---|---|---|
| Enhancing the Incorporation of Crop Residues into Soil | Careful Selection of the Most Resilient Crops | Flexible Rotations |
| Attention to Planting Times | Mixed Crops | Choice of Irrigation Timing and Technology |
| Agroforestry | Insurance of Crops | Diversified Livestock Farms |
| Risk management tools adoption | Assignment to Institutional Development Programs | Land Lease |
| Supplementary Irrigation | Use of Technologies: Sensors And Digital Mapping, Remote Sensing, Meteorological Recording Station | Reduction Of Groundwater Withdrawals. |
| Alteration of Plant Population Density | Stubble Retention | Improved Weed Control |

Several types of potential interventions emerge from Table 3. Some of these are more technical, such as the incorporation of crop residues into the soil [56,57], while others have different natures, such as risk management instruments [10,58]. For example, financial tools, such as crop insurance, allow transferring risks to other subjects, helping farmers face the increasing vulnerability of their activity [59]. Some countries are introducing such instruments in their policies supporting the agricultural sector; these tools are accepted by the WTO because they meet the criterion of causing the least distortions [60].

There are also some interventions that appear to be more economic, such as better weed management, while others seem very expensive and structurally impactful, such as switching to a more resilient crop or moving to a long-term vision based on, for example, an agroforestry approach [61].

The high importance given to technology also emerges. The use of technology (e.g., remote sensing or meteorological recording stations) can greatly support farmers' work and limit their level of uncertainty [62,63].

Developing and implementing new strategies to reduce the impacts of climate change requires implementing information generation and sharing. Indeed, given the vulnerability of the studied areas, increased data and data sharing are necessary to assist land managers and policymakers in understanding environmental dynamics [64].

3.1.3. Blue Cluster: The Resilience of Arid Areas

As in the Green cluster, the Blue cluster includes 16 articles. The themes that emerged from this cluster can be summarised as follows: (i) soil analysis in arid and semi-arid areas and (ii) the resilience of arid areas.

As shown in Figure 5, the Blue cluster is very far from the Green cluster. In fact, it contains no papers analysing potential approaches to respond to climate change. On the other hand, the use of technological equipment emerges as an innovative strategy to respond to new challenges [60,65]. However, Figure 5 shows the proximity of the Blue cluster to the Red cluster: A focus on aspects such as inclusion and vulnerability of drylands emerges in both cases [66].

As already anticipated, the literature [67–69] shows the importance of analysing the soil of arid and semi-arid areas. One of the indicators that appear appropriate is soil organic carbon, which is used to monitor soil degradation and desertification processes in arid areas [70].

This cluster points to the need to analyse these areas' soil through quantitative approaches. This is partly because unsustainable land use is considered the main negative factor in the degradation of drylands [71]. On the other hand, soil management

analysis is seen as a multidisciplinary challenge that includes social, economic, and political factors [68,70].

Many scholars [72,73] have stressed the need to analyse the resilience of these areas. Starting from the definition of resilience (an area's ability to resist disturbances and maintain overall functioning), this turns out to be a challenge to be analysed by researchers and stakeholders. Resilience has been incorporated within many of the United Nations Sustainable Development Goals, but the recent attention given to this issue brings with it the difficulty of measurement, especially with reference to the territory over time and space [74]. In fact, although several resilience measurements have been proposed (e.g., resilience to disturbance, rate of recovery from disturbance, and robustness), it remains difficult to find a strategy to make them applicable and operational [72,73].

### 3.1.4. Yellow Cluster: A Variety of Indices

The Yellow cluster highlights a very important thematic in dryland analysis: the variety of indices used in the literature.

Many authors have used or implemented several indices to analyse issues related to the management of current challenges and the implementation of econometric models.

According to Liang et al. [75], some commonly used indices include the Palmer Drought Severity Index, Standardised Precipitation Index, and Standardised Precipitation Evapotranspiration Index.

Other indices analyse the phenological timing and bioclimatic requirements of certain tree species, monitoring soils' physical, chemical, and biological status to manage their sustainability, such as the Thornthwaite moisture index [76].

A growing body of literature appears to be focused on analysing the issue using this wide choice of indices, but the topic remains very controversial. It is important to find data, but the availability and retrieval of data turn out to be difficult in such contiguous areas [75].

On the other hand, the accuracy and reliability of calculating the indices must be contextualised to the areas where the data are collected. In fact, these data might suggest the implementation of strategies that cannot be adopted or are not effective because of non-quantifiable aspects of the problem, such as farmers' tacit knowledge or their readiness to respond to changes [77].

### 3.1.5. Purple Cluster: The Multifunctionality of Agricultural Systems

The Purple cluster contains the least number of documents (6). As shown in Figure 5, the last cluster has many relationships with the Green cluster. The main synergies are related to the focus on possible strategies to mitigate the effects of climate change on agriculture.

Several authors (such as [78]) have highlighted the need to select crop varieties that are more appropriate to the current environmental situation and are able to withstand difficult conditions. For example, Acharya et al. [79] reported on the cactus pear (Opuntia ficus-indica), a crop famous for its food and medicinal values. This crop has low water requirements and is tolerant to high temperatures, making it an advantageous crop for growing in arid and semi-arid environments [79].

Other strategies that have emerged from the literature include significant useful water-saving adaptations, e.g., improvement of crop varieties, mechanisation of some agricultural steps, improvement of agricultural infrastructure, agricultural subsidy, and development of dry farming technologies [56,80,81]. On the other hand, the analysis revealed various actions that can increase the efficiency of precipitation management, such as responsible fertilisation, alteration in crop rotation, or inclusion of annual forage crops [81,82]

Two interconnected themes also emerge from the Purple cluster that are very important to the issue: the management of agricultural systems and the management of common goods.

According to Wang et al. [82], agricultural systems are defined as multifunctional ecosystems. Their function is not only limited to food production but also to the provision

of ecosystem services. In addition, various natural resource management issues require interdisciplinary knowledge.

This view of the topic relates to another paper in this cluster written by Loch and Gregg [81], who analysed the difficult management of common goods, such as water quality. Specifically, public institutions generally invest in two main activities: the first is represented by administering, monitoring, and enforcing current policy provisions, and the second focuses on designing and transitioning existing management agreements or creating new ones.

Hence, this cluster is focused on sustainable, long-term management of agricultural systems, which must be profitable and less impactful on natural resources. This opens the way to a new challenge addressed to policy makers and researchers: Even where the data appear to be fragmented, there is an emerging need to manage ecosystems to make them resilient without overlooking the key characteristic of these areas, which is multifunctionality.

## 4. Discussion

Results show researchers' increasing focus on issues linking drylands and climate change (Figure 2). In addition, five clusters emerged that divided the current literature into five macro areas.

More specifically, the need to include farmers and develop increasingly participatory policy interventions emerged (Red cluster). Secondly (Green cluster), the more technical papers analysed potential adaptive interventions implemented by farmers and supported by policies. Subsequently (Blue cluster), the literature has emphasised the importance of a relatively new concept: the resilience of territories and the difficulty of finding indices to measure it. At the same time, the Yellow cluster highlighted the presence of a multitude of indices used for the analysis of these issues. Finally, the Purple cluster reported the need to analyse agricultural systems by considering their multifunctionality.

In fact, the analysis revealed a surprising aspect of the management of this issue. When considering drylands as significantly vulnerable areas, the literature suggests using participatory approaches intended to increase the social capital of the area significantly and aiming to construct territorial resilience [52]. In fact, social capital influences farmers' ability to adapt to new threats, and the process is facilitated by the increased availability of information, greater trust in institutions, and facilitated use of resources [44,83,84]. On the other hand, promoting the resilience of areas is necessary to mitigate the effects of climate change and re-evaluate affected areas [72,73].

Indeed, resilience concerns unpredictable events and their impacts, suggesting the necessity of strategies to enhance systems' adaptability and transformability [2]. Climate change as a process is not expected to retreat to the extent that the world will not need to adapt to its effects. Another key factor in fighting climate change's impacts is social capital, as knowing about specific topics in agricultural areas could lead to the implementation of winning strategies. Moreover, resource scarcity limiting the development of the agricultural sector poses the need for new, developing factors, such as those related to resilience and social capital. For example, local development could be enhanced by the increasing sharing of knowledge, implemented thanks to higher levels of social capital. The world cannot continue to face the economic impact of climate change as it has always done. Social capital is the key factor in generating resilience since a territory's adaptive capacity depends on local knowledge, networks, and attitude towards innovation.

Another strategy that appears to be under-adopted is the use of financial tools to transfer risks. In fact, many authors (e.g., [85,86]) have reported that financial insurance for extreme events can be used to face the impacts of climate change and could be an important adaptation strategy. Agricultural risk management is becoming increasingly important, as seen, for example, in the latest European Common Agricultural Policy 2014–2020 (CAP) [87,88]. The Common Agricultural Policy (Reg UE 1305/2013) provides various instruments for risk management (art. 36)—i.e., agricultural insurance (art. 37),

mutual funds (art. 38), and the income stabilisation tool (art. 39) [21]. However, the demand for these tools is still low, and integrated policy intervention must be implemented to enhance their adoption [89]. To this end, it is also important to have a large set of data to better structure adequate policies and to give each area a specific instrument to promote local development. From this review, the need to use data and information also emerged: even if it is a difficult challenge and there are many obstacles to overcome, the first step to supporting territories is knowing those territories.

## 5. Conclusions

In recent years, the impacts of climate change have received significant attention in literature and in many policy documents [42]. More specifically, increasing temperatures and the constantly changing magnitude and frequency of meteorological events are causing growing concerns about new strategies to react to these threats, which will be more impactful in the growing arid and semi-arid areas [12,13].

To support the policy makers' decision making and help develop cross-sectional strategies to support farmers, this paper is intended to examine how this issue has been analysed in the literature and to highlight the opportunities and bottlenecks on which interventions now needed can be based. The outcome was achieved by applying a mixed method based on a systematic literature review and subsequent bibliometric analysis using VOSviewer software.

The literature review revealed the relevance of the analysed issue. Preliminary results showed 250 articles related to dryland farming and climate change. After the screening, only 79 articles were selected to be included in this review. Of these works, more than 80% were published after 2015, when Agenda 2030 was signed, implying an increase in researchers' efforts to study these phenomena to find solutions to climate change and giving major attention to the role of the agricultural sector.

The review also exposed the multidisciplinary nature of the investigated issue, emerging from the cluster division of the works. Papers were allocated in five clusters, sometimes overlapping. The cluster of greater relevance concerns farmer inclusion and policy interventions (21 papers out of 79), followed by the potential adaptations to climate change and the resilience of arid areas (each containing 16 works). The literature focused on the agricultural sector's weaknesses to define a common policy strategy.

In fact, this issue is even more accentuated in the primary sector, which shows a close connection and dependence between the environmental threats and the economic stability of the whole sector [7]. Indeed, yield productivity increases and evolves into the augmented volatility of farmers' revenue. To face these issues in the current scenario, focused mainly on practical solutions, it is also necessary to integrate several instruments (e.g., of agronomic and financial natures), thanks to the implementation of adequate policies.

## 6. Policy Implications

As highlighted in the Introduction section, this study aims to fill the literature gap about policy implementation since most of the previous studies have a technical approach rather than a policy one. As it emerged from the results' discussion, there is an increasing need to develop and adopt participatory policy interventions. This is particularly true for the social capital that was indicated as one of the fundamental elements of improving the effectiveness and reducing the risk of failure of the policies. To this aim, a crucial role will be played by the social and human capital as well as by generational renewal. Therefore, training instruments and access to information for farmers will represent elements of crucial importance that could help to build and boost social capital and trust in institutions, facilitating the resources use.

Moreover, a crucial role in guiding farmers through risk management and facing climate change strategies will be played by institutions. Specifically, they could monitor the changes in drought levels and promote coping, as well as implement multidisciplinary strategies by integrating different actors.

A further important contribution that institutions may provide is also represented by investments in innovations. In fact, for some farmers, especially the smallest ones, it is hard to invest in the specific tools and technologies needed to face climate change. Therefore, policies should promote technological renewal and help the farmers to adopt innovations by providing financial support and fostering knowledge sharing.

It is also crucial to analyse agricultural systems, considering their multifunctionality, to build up territorial resilience and sustainable development. In fact, when income from production decreases, activities connected to the rural reality (for example, agritourism) play a relevant role. In this direction, adopting local crop varieties could be a winning strategy to foster agricultural sustainability in marginal areas based on their ability to exploit limited resources and cope with extreme weather conditions.

Another way to address the difficulties caused by climate change and by territories' weaknesses is represented by financial tools to transfer risks on the market. Even if they are largely under-adopted, they could represent a relevant chance to promote the economic sustainability of farmers' activities. Policies should promote their use not only by providing financial support (fundamental for their desirability) but also by facilitating the availability of such subsidies. To enhance this sector (and many others), it is also important to facilitate access to information that can help farmers and insurers gain a realistic idea of activities' riskiness.

Finally, access to data would help farmers implement precision agriculture, another possible strategy to maintain agricultural activities in marginal areas facing adverse climate conditions. Furthermore, access to meteorological data is essential to plan an efficient cropping calendar and help farmers determine the right climatic conditions for crop growing.

## 7. Limitations and Directions for Future Research

This paragraph contains a discussion of some limitations of the present study, followed by some suggestions for possible future research.

The idea of this paper was to analyse the issue of climate change and dryland farming from the perspective of a global challenge. The main limit of the study is that other scholars might have chosen other keywords on which to base their investigation of the current situation in this field. However, in this case, the goal was to highlight future paths on which to base cross-sectional effective and efficient policy interventions. Due to the literature gap and the increasing attention given to the issue by institutional policies, future researchers may analyse the effects of climate change on the agricultural and agricultural risk-management sectors, particularly in developed countries. Moreover, research on separate topics could be carried out to catch up on other relevant aspects.

It is also noteworthy that the issues of drought and desertification are somewhat different in each geographical area. Therefore, evaluating how the problem has been analysed so far in various countries could be of interest. This would also be interesting considering countries' strengths, as they have different types of power to face climate change's effects. Poorer territories have fewer economic resources to invest in providing policy instruments for farmers and, often, less human capital to implement innovative strategies. For this reason, broad studies (such as the present one) represent elements of interest that allow knowledge spread among territories.

**Author Contributions:** Conceptualisation, N.d.S. and R.S.; methodology, N.d.S.; formal analysis, N.d.S.; investigation, R.S. and N.d.S.; writing—original draft preparation, I.R. and N.d.S.; writing—review and editing, R.S., N.d.S. and I.R. All authors have read and agreed to the published version of the manuscript.

**Funding:** This research received no external funding.

**Data Availability Statement:** Informed consent was obtained from all subjects involved in the study.

**Conflicts of Interest:** The authors declare no conflict of interest.

## Appendix A

**Table A1.** Major journals publishing studies on this topic.

| Journal | Number of Papers |
| --- | --- |
| Climatic Change | 6 |
| Sustainability (Switzerland) | 6 |
| Nature Communications | 4 |
| Agricultural Systems | 3 |
| Ecology and Society | 3 |
| Environmental Research Letters | 3 |
| Land Use Policy | 3 |
| Agriculture, Ecosystems and Environment | 2 |
| Current Opinion in Environmental Sustainability | 2 |
| International Journal of Water Resources Development | 2 |
| Journal of Ecology | 2 |
| Land Degradation and Development | 2 |
| Science of the Total Environment | 2 |
| Water (Switzerland) | 2 |
| Arid Land Research and Management | 1 |
| Biogeochemistry | 1 |
| Biology Letters | 1 |
| Carbon Balance and Management | 1 |
| Climate and Development | 1 |
| Diversity | 1 |
| Ecological Indicators | 1 |
| Ecosystems | 1 |
| Environmental Management | 1 |
| European Journal of Soil Science | 1 |
| Field Crops Research | 1 |
| Forests | 1 |
| Forests Trees and Livelihoods | 1 |
| Frontiers in Ecology and The Environment | 1 |
| Gcb Bioenergy | 1 |
| Geoforum | 1 |
| Heliyon | 1 |
| Ids Bulletin | 1 |
| International Journal of Agricultural Sustainability | 1 |
| International Journal of Environmental Research and Public Health | 1 |
| Journal of Applied Ecology | 1 |
| Journal of Environmental Management | 1 |
| New Phytologist | 1 |
| Njas - Wageningen Journal of Life Sciences | 1 |
| Remote Sensing of Environment | 1 |
| Science | 1 |
| Soil Biology and Biochemistry | 1 |
| Theoretical and Applied Climatology | 1 |
| Water Sa | 1 |

**Table A2.** Papers considered in the study.

| VOS Label | Authors | Title | Journal | Year | Cluster | Citations | Norm. Citations |
|---|---|---|---|---|---|---|---|
| **Abrouk M. (2020)** | Abrouk M.; Ahmed H.I.; Cubry P.; Šimoníková D.; Cauet S.; Pailles Y.; Bettgenhaeuser J.; Gapa L.; Scarcelli N.; Couderc M.; Zekraoui L.; Kathiresan N.; Čížková J.; Hřibová E.; Doležel J.; Arribat S.; Bergès H.; Wieringa J.J.; Gueye M.; Kane N.A.; Leclerc C.; Causse S.; Vancoppenolle S.; Billot C.; Wicker T.; Vigouroux Y.; Barnaud A.; Krattinger S.G. | Fonio Millet Genome Unlocks African Orphan Crop Diversity For Agriculture In A Changing Climate | Nature Communications, 11(1) | 2020 | Red | 26 | 15.089 |
| **Young G. (2009)** | Young G.; Zavala H.; Wandel J.; Smit B.; Salas S.; Jimenez E.; Fiebig M.; Espinoza R.; Diaz H.; Cepeda J. | Vulnerability And Adaptation In A Dryland Community Of The Elqui Valley, Chile | Climatic Change, 98(1–2), 245–276 | 2009 | Red | 65 | 14.943 |
| **Manaye A. (2021)** | Manaye A.; Tesfamariam B.; Tesfaye M.; Worku A.; Gufi Y. | Tree Diversity And Carbon Stocks In Agroforestry Systems In Northern Ethiopia | Carbon Balance And Management, 16(1) | 2021 | Red | 8 | 13.521 |
| **Villamor G.B. (2016)** | Villamor G.B.; Badmos B.K. | Grazing Game: A Learning Tool For Adaptive Management In Response To Climate Variability In Semiarid Areas Of Ghana | Ecology And Society, 21(1) | 2016 | Red | 28 | 12.174 |
| **Endale Y. (2017)** | Endale Y.; Derero A.; Argaw M.; Muthuri C. | Farmland Tree Species Diversity And Spatial Distribution Pattern In Semi-Arid East Shewa, Ethiopia | Forests Trees And Livelihoods, 26(3), 199–214 | 2017 | Red | 32 | 12.075 |
| **Moreno-Jiménez E. (2022)** | Moreno-Jiménez E.; Orgiazzi A.; Jones A.; Saiz H.; Aceña-Heras S.; Plaza C. | Aridity And Geochemical Drivers Of Soil Micronutrient And Contaminant Availability In European Drylands | European Journal Of Soil Science, 73(1) | 2022 | Red | 1 | 11.667 |
| **Kattumuri R. (2017)** | Kattumuri R.; Ravindranath D.; Esteves T. | Local Adaptation Strategies In Semi-Arid Regions: Study Of Two Villages In Karnataka, India | Climate And Development, 9(1), 36–49 | 2017 | Red | 27 | 10.189 |
| **Tui S.H.-K. (2021)** | Tui S.H.-K.; Descheemaeker K.; Valdivia R.O.; Masikati P.; Sisito G.; Moyo E.N.; Crespo O.; Ruane A.C.; Rosenzweig C. | Climate Change Impacts And Adaptation For Dryland Farming Systems In Zimbabwe: A Stakeholder-Driven Integrated Multi-Model Assessment | Climatic Change, 168(1–2) | 2021 | Red | 6 | 10.141 |
| **Scoones I. (2004)** | Scoones I. | Climate Change And The Challenge Of Non-Equilibrium Thinking | Ids Bulletin, 35(3), 114–119 | 2004 | Red | 29 | 1 |
| **Weston P. (2015)** | Weston P.; Hong R.; Kaboré C.; Kull C.A. | Farmer-Managed Natural Regeneration Enhances Rural Livelihoods In Dryland West Africa | Environmental Management, 55(6), 1402–1417 | 2015 | Red | 50 | 0.9363 |

**Table A2.** *Cont.*

| VOS Label | Authors | Title | Journal | Year | Cluster | Citations | Norm. Citations |
|---|---|---|---|---|---|---|---|
| **Antwi-Agyei P. (2015)** | Antwi-Agyei P.; Dougill A.J.; Stringer L.C. | Impacts Of Land Tenure Arrangements On The Adaptive Capacity Of Marginalized Groups: The Case Of Ghana's Ejura Sekyedumase And Bongo Districts | Land Use Policy, 49, 203–212 | 2015 | Red | 41 | 0.7678 |
| **Sun Y. (2021)** | Sun Y.; Sun Y.; Yao S.; Akram M.A.; Hu W.; Dong L.; Li H.; Wei M.; Gong H.; Xie S.; Aqeel M.; Ran J.; Degen A.A.; Guo Q.; Deng J. | Impact Of Climate Change On Plant Species Richness Across Drylands In China: From Past To Present And Into The Future | Ecological Indicators, 132 | 2021 | Red | 4 | 0.6761 |
| **Recha J.W. (2016)** | Recha J.W.; Mati B.M.; Nyasimi M.; Kimeli P.K.; Kinyangi J.M.; Radeny M. | Changing Rainfall Patterns And Farmers' Adaptation Through Soil Water Management Practices In Semi-Arid Eastern Kenya | Arid Land Research And Management, 30(3), 229–238 | 2016 | Red | 15 | 0.6522 |
| **Kalame F.B. (2011)** | Kalame F.B.; Luukkanen O.; Kanninen M. | Making The National Adaptation Programme Of Action (Napa) More Responsive To The Livelihood Needs Of Tree Planting Farmers, Drawing On Previous Experience In Dryland Sudan | Forests, 2(4), 948–960 | 2011 | Red | 5 | 0.4762 |
| **Ndhleve S. (2017)** | Ndhleve S.; Nakin M.D.V.; Longo-Mbenza B. | Impacts Of Supplemental Irrigation As A Climate Change Adaptation Strategy For Maize Production: A Case Of The Eastern Cape Province Of South Africa | Water Sa, 43(2), 222–228 | 2017 | Red | 12 | 0.4528 |
| **Galvin K.A. (2021)** | Galvin K.A. | Transformational Adaptation In Drylands | Current Opinion In Environmental Sustainability, 50, 64–71 | 2021 | Red | 2 | 0.338 |
| **Arficho M. (2020)** | Arficho M.; Thiel A. | Does Land-Use Policy Moderate Impacts Of Climate Anomalies On Lulc Change In Dry-Lands? An Empirical Enquiry Into Drivers And Moderators Of Lulc Change In Southern Ethiopia | Sustainability (Switzerland), 12(15) | 2020 | Red | 4 | 0.2321 |
| **Mulligan M. (2015)** | Mulligan M. | Climate Change And Food-Water Supply From Africa's Drylands: Local Impacts And Teleconnections Through Global Commodity Flows | International Journal Of Water Resources Development, 31(3), 450–460 | 2015 | Red | 5 | 0.0936 |

**Table A2.** *Cont.*

| VOS Label | Authors | Title | Journal | Year | Cluster | Citations | Norm. Citations |
|---|---|---|---|---|---|---|---|
| **Gebru K.M. (2020)** | Gebru K.M.; Woldearegay K.; Van Steenbergen F.; Beyene A.; Vera L.F.; Gebreegziabher K.T.; Alemayhu T. | Adoption Of Road Water Harvesting Practices And Their Impacts: Evidence From A Semi-Arid Region Of Ethiopia | Sustainability (Switzerland), 12(21), 1–25 | 2020 | Red | 1 | 0.058 |
| **Chanza N. (2022)** | Chanza N.; Musakwa W. | Ecological And Hydrological Indicators Of Climate Change Observed By Dryland Communities Of Malipati In Chiredzi, Zimbabwe | Diversity, 14(7) | 2022 | Red | 0 | 0 |
| **Samuel J. (2022)** | Samuel J.; Rao C.A.R.; Raju B.M.K.; Reddy A.A.; Pushpanjali, Reddy A.G.K.; Kumar R.N.; Osman M.; Singh V.K.; Prasad J.V.N.S. | Assessing The Impact Of Climate Resilient Technologies In Minimizing Drought Impacts On Farm Incomes In Drylands | Sustainability (Switzerland), 14(1) | 2022 | Red | 0 | 0 |
| **Hassani A. (2021)** | Hassani A.; Azapagic A.; Shokri N. | Global Predictions Of Primary Soil Salinization Under Changing Climate In The 21st Century | Nature Communications, 12(1) | 2021 | Green | 18 | 30.423 |
| **Cotterman K.A. (2018)** | Cotterman K.A.; Kendall A.D.; Basso B.; Hyndman D.W. | Groundwater Depletion And Climate Change: Future Prospects Of Crop Production In The Central High Plains Aquifer | Climatic Change, 146(1–2), 187–200 | 2018 | Green | 43 | 22.114 |
| **Macfadyen S. (2018)** | Macfadyen S.; Mcdonald G.; Hill M.P. | From Species Distributions To Climate Change Adaptation: Knowledge Gaps In Managing Invertebrate Pests In Broad-Acre Grain Crops | Agriculture, Ecosystems And Environment, 253, 208–219 | 2018 | Green | 34 | 17.486 |
| **Tack J. (2017)** | Tack J.; Barkley A.; Hendricks N. | Irrigation Offsets Wheat Yield Reductions From Warming Temperatures | Environmental Research Letters, 12(11) | 2017 | Green | 45 | 16.981 |
| **Stöckle C.O. (2018)** | Stöckle C.O.; Higgins S.; Nelson R.; Abatzoglou J.; Huggins D.; Pan W.; Karimi T.; Antle J.; Eigenbrode S.D.; Brooks E. | Evaluating Opportunities For An Increased Role Of Winter Crops As Adaptation To Climate Change In Dryland Cropping Systems Of The U.S. Inland Pacific Northwest | Climatic Change, 146(1–2), 247–261 | 2018 | Green | 33 | 16.971 |
| **Thamo T. (2017)** | Thamo T.; Addai D.; Pannell D.J.; Robertson M.J.; Thomas D.T.; Young J.M. | Climate Change Impacts And Farm-Level Adaptation: Economic Analysis Of A Mixed Cropping–Livestock System | Agricultural Systems, 150, 99–108 | 2017 | Green | 41 | 15.472 |
| **Luo Y. (2020)** | Luo Y.; Zhang Z.; Li Z.; Chen Y.; Chen Y.; Zhang L.; Cao J.; Tao F.; Tao F. | Identifying The Spatiotemporal Changes Of Annual Harvesting Areas For Three Staple Crops In China By Integrating Multi-Data Sources | Environmental Research Letters, 15(7) | 2020 | Green | 25 | 14.509 |

**Table A2.** *Cont.*

| VOS Label | Authors | Title | Journal | Year | Cluster | Citations | Norm. Citations |
|---|---|---|---|---|---|---|---|
| **Yang C. (2019)** | Yang C.; Fraga H.; Van Ieperen W.; Trindade H.; Santos J.A. | Effects Of Climate Change And Adaptation Options On Winter Wheat Yield Under Rainfed Mediterranean Conditions In Southern Portugal | Climatic Change, 154(1–2), 159–178 | 2019 | Green | 39 | 12.857 |
| **Ghahramani A. (2016)** | Ghahramani A.; Moore A.D. | Impact Of Climate Changes On Existing Crop-Livestock Farming Systems | Agricultural Systems, 146, 142–155 | 2016 | Green | 26 | 11.304 |
| **Karimi T. (2021)** | Karimi T.; Stöckle C.O.; Higgins S.S.; Nelson R.L. | Impact Of Climate Change On Greenhouse Gas Emissions And Water Balance In A Dryland-Cropping Region With Variable Precipitation | Journal Of Environmental Management, 287 | 2021 | Green | 6 | 10.141 |
| **Asseng S. (2013)** | Asseng S.; Pannell D.J. | Adapting Dryland Agriculture To Climate Change: Farming Implications And Research And Development Needs In Western Australia | Climatic Change, 118(2), 167–181 | 2013 | Green | 53 | 1 |
| **Monjardino M. (2010)** | Monjardino M.; Revell D.; Pannell D.J. | The Potential Contribution Of Forage Shrubs To Economic Returns And Environmental Management In Australian Dryland Agricultural Systems | Agricultural Systems, 103(4), 187–197 | 2010 | Green | 39 | 1 |
| **Shayanmehr S. (2020)** | Shayanmehr S.; Henneberry S.R.; Sabouni M.S.; Foroushani N.S. | Drought, Climate Change, And Dryland Wheat Yield Response: An Econometric Approach | International Journal Of Environmental Research And Public Health, 17(14), 1–18 | 2020 | Green | 14 | 0.8125 |
| **Antle J.M. (2018)** | Antle J.M.; Zhang H.; Mu J.E.; Abatzoglou J.; Stöckle C. | Methods To Assess Between-System Adaptations To Climate Change: Dryland Wheat Systems In The Pacific Northwest United States | Agriculture, Ecosystems And Environment, 253, 195–207 | 2018 | Green | 15 | 0.7714 |
| **Lawrence P.G. (2018)** | Lawrence P.G.; Maxwell B.D.; Rew L.J.; Ellis C.; Bekkerman A. | Vulnerability Of Dryland Agricultural Regimes To Economic And Climatic Change | Ecology And Society, 23(1) | 2018 | Green | 13 | 0.6686 |
| **Zhang H. (2018)** | Zhang H.; Mu J.E.; Mccarl B.A. | Adaptation To Climate Change Via Adjustment In Land Leasing: Evidence From Dryland Wheat Farms In The U.S. Pacific Northwest | Land Use Policy, 79, 424–432 | 2018 | Green | 3 | 0.1543 |

**Table A2.** *Cont.*

| VOS Label | Authors | Title | Journal | Year | Cluster | Citations | Norm. Citations |
|---|---|---|---|---|---|---|---|
| **Burrell A.L. (2020)** | Burrell A.L.; Evans J.P.; De Kauwe M.G. | Anthropogenic Climate Change Has Driven Over 5 Million Km2 Of Drylands Towards Desertification | Nature Communications, 11(1) | 2020 | Blue | 77 | 44.688 |
| **Smith W.K. (2019)** | Smith W.K.; Dannenberg M.P.; Yan D.; Herrmann S.; Barnes M.L.; Barron-Gafford G.A.; Biederman J.A.; Ferrenberg S.; Fox A.M.; Hudson A.; Knowles J.F.; Macbean N.; Moore D.J.P.; Nagler P.L.; Reed S.C.; Rutherford W.A.; Scott R.L.; Wang X.; Yang J. | Remote Sensing Of Dryland Ecosystem Structure And Function: Progress, Challenges, And Opportunities | Remote Sensing Of Environment, 233 | 2019 | Blue | 107 | 35.275 |
| **Bonachela J.A. (2015)** | Bonachela J.A.; Pringle R.M.; Sheffer E.; Coverdale T.C.; Guyton J.A.; Caylor K.K.; Levin S.A.; Tarnita C.E. | Termite Mounds Can Increase The Robustness Of Dryland Ecosystems To Climatic Change | Science, 347(6222), 651–655 | 2015 | Blue | 147 | 27.528 |
| **Twyman C. (2011)** | Twyman C.; Fraser E.D.G.; Stringer L.C.; Quinn C.; Dougill A.J.; Ravera F.; Crane T.A.; Sallu S.M. | Climate Science, Development Practice, And Policy Interactions In Dryland Agroecological Systems | Ecology And Society, 16(3), 19 | 2011 | Blue | 16 | 15.238 |
| **Kusch E. (2022)** | Kusch E.; Davy R.; Seddon A.W.R. | Vegetation-Memory Effects And Their Association With Vegetation Resilience In Global Drylands | Journal Of Ecology, 110(7), 1561–1574 | 2022 | Blue | 1 | 11.667 |
| **Cai L. (2022)** | Cai L.; Wang H.; Liu Y.; Fan D.; Li X. | Is Potential Cultivated Land Expanding Or Shrinking In The Dryland Of China? Spatiotemporal Evaluation Based On Remote Sensing And Svm | Land Use Policy, 112 | 2022 | Blue | 1 | 11.667 |
| **He L. (2021)** | He L.; Li Z.-L.; Wang X.; Xie Y.; Ye J.-S. | Lagged Precipitation Effect On Plant Productivity Is Influenced Collectively By Climate And Edaphic Factors In Drylands | Science Of The Total Environment, 755 | 2021 | Blue | 6 | 10.141 |
| **Gaitán J.J. (2014)** | Gaitán J.J.; Bran D.; Oliva G.; Maestre F.T.; Aguiar M.R.; Jobbagy E.; Buono G.; Ferrante D.; Nakamatsu V.; Ciari G.; Salomone J.; Massara V. | Plant Species Richness And Shrub Cover Attenuate Drought Effects On Ecosystem Functioning Across Patagonian Rangelands | Biology Letters, 10(10) | 2014 | Blue | 17 | 1 |
| **Pervez Bharucha Z. (2021)** | Pervez Bharucha Z.; Attwood S.; Badiger S.; Balamatti A.; Bawden R.; Bentley J.W.; Chander M.; Davies L.; Dixon H.; Dixon J.; D'souza M.; Butler Flora C.; Green M.; Joshi D.; Komarek A.M.; Ruth Mcdermid L.; Mathijs E.; Rola A.C.; Patnaik S.; Pattanayak S.; Pingali P.; Vara Prasad V.P.; Rabbinge R.; Ramanjaneyulu G.V.; Ravindranath N.H.; Sage C.; Saha A.; Salvatore C.; Patnaik Saxena L.; Singh C.; Smith P.; Srinidhi A.; Sugam R.; Thomas R.; Uphoff N.; Pretty J. | The Top 100 Questions For The Sustainable Intensification Of Agriculture In India's Rainfed Drylands | International Journal Of Agricultural Sustainability, 19(2), 106–127 | 2021 | Blue | 4 | 0.6761 |

**Table A2.** *Cont.*

| VOS Label | Authors | Title | Journal | Year | Cluster | Citations | Norm. Citations |
|---|---|---|---|---|---|---|---|
| **Gaitán J.J. (2019)** | Gaitán J.J.; Maestre F.T.; Bran D.E.; Buono G.G.; Dougill A.J.; García Martínez G.; Ferrante D.; Guuroh R.T.; Linstädter A.; Massara V.; Thomas A.D.; Oliva G.E. | Biotic And Abiotic Drivers Of Topsoil Organic Carbon Concentration In Drylands Have Similar Effects At Regional And Global Scales | Ecosystems, 22(7), 1445–1456 | 2019 | Blue | 15 | 0.4945 |
| **Browning D.M. (2015)** | Browning D.M.; Rango A.; Karl J.W.; Laney C.M.; Vivoni E.R.; Tweedie C.E. | Emerging Technological And Cultural Shifts Advancing Drylands Research And Management | Frontiers In Ecology And The Environment, 13(1), 52–60 | 2015 | Blue | 24 | 0.4494 |
| **Kwiecinski J.V. (2020)** | Kwiecinski J.V.; Stricker E.; Sinsabaugh R.L.; Collins S.L. | Rainfall Pulses Increased Short-Term Biocrust Chlorophyll But Not Fungal Abundance Or N Availability In A Long-Term Dryland Rainfall Manipulation Experiment | Soil Biology And Biochemistry, 142 | 2020 | Blue | 7 | 0.4062 |
| **Stavi I. (2021)** | Stavi I.; Yizhaq H.; Szitenberg A.; Zaady E. | Patch-Scale To Hillslope-Scale Geodiversity Alleviates Susceptibility Of Dryland Ecosystems To Climate Change: Insights From The Israeli Negev | Current Opinion In Environmental Sustainability, 50, 129–137 | 2021 | Blue | 2 | 0.338 |
| **Schreiner-Mcgraw A.P. (2019)** | Schreiner-Mcgraw A.P.; Ajami H.; Vivoni E.R. | Extreme Weather Events And Transmission Losses In Arid Streams | Environmental Research Letters, 14(8) | 2019 | Blue | 8 | 0.2637 |
| **Bharucha Z.P. (2019)** | Bharucha Z.P. | This Is What Nature Has Become: Tracing Climate And Water Narratives In India's Rainfed Drylands | Geoforum, 101, 285–293 | 2019 | Blue | 7 | 0.2308 |
| **Zhang Y. (2022)** | Zhang Y.; Gentine P.; Luo X.; Lian X.; Liu Y.; Zhou S.; Michalak A.M.; Sun W.; Fisher J.B.; Piao S.; Keenan T.F. | Increasing Sensitivity Of Dryland Vegetation Greenness To Precipitation Due To Rising Atmospheric Co2 | Nature Communications, 13(1) | 2022 | Blue | 0 | 0 |
| **Delgado-Baquerizo M. (2018)** | Delgado-Baquerizo M.; Maestre F.T.; Eldridge D.J.; Bowker M.A.; Jeffries T.C.; Singh B.K. | Biocrust-Forming Mosses Mitigate The Impact Of Aridity On Soil Microbial Communities In Drylands: Observational Evidence From Three Continents | New Phytologist, 220(3), 824–835 | 2018 | Yellow | 30 | 15.429 |
| **Baldauf S. (2021)** | Baldauf S.; Porada P.; Raggio J.; Maestre F.T.; Tietjen B. | Relative Humidity Predominantly Determines Long-Term Biocrust-Forming Lichen Cover In Drylands Under Climate Change | Journal Of Ecology, 109(3), 1370–1385 | 2021 | Yellow | 8 | 13.521 |

**Table A2.** *Cont.*

| VOS Label | Authors | Title | Journal | Year | Cluster | Citations | Norm. Citations |
|---|---|---|---|---|---|---|---|
| **Eldridge D.J. (2020)** | Eldridge D.J.; Delgado-Baquerizo M.; Quero J.L.; Ochoa V.; Gozalo B.; García-Palacios P.; Escolar C.; García-Gómez M.; Prina A.; Bowker M.A.; Bran D.E.; Castro I.; Cea A.; Derak M.; Espinosa C.I.; Florentino A.; Gaitán J.J.; Gatica G.; Gómez-González S.; Ghiloufi W.; Gutierrez J.R.; Gusmán-Montalván E.; Hernández R.M.; Hughes F.M.; Muiño W.; Monerris J.; Ospina A.; Ramírez D.A.; Ribas-Fernández Y.A.; Romão R.L.; Torres-Díaz C.; Koen T.B.; Maestre F.T. | Surface Indicators Are Correlated With Soil Multifunctionality In Global Drylands | Journal Of Applied Ecology, 57(2), 424–435 | 2020 | Yellow | 21 | 12.188 |
| **Krichels A.H. (2022)** | Krichels A.H.; Homyak P.M.; Aronson E.L.; Sickman J.O.; Botthoff J.; Shulman H.; Piper S.; Andrews H.M.; Jenerette G.D. | Rapid Nitrate Reduction Produces Pulsed No And N2o Emissions Following Wetting Of Dryland Soils | Biogeochemistry, 158(2), 233–250 | 2022 | Yellow | 3 | 3.5 |
| **Delgado-Baquerizo M. (2020)** | Delgado-Baquerizo M.; Doulcier G.; Eldridge D.J.; Stouffer D.B.; Maestre F.T.; Wang J.; Powell J.R.; Jeffries T.C.; Singh B.K. | Increases In Aridity Lead To Drastic Shifts In The Assembly Of Dryland Complex Microbial Networks | Land Degradation And Development, 31(3), 346–355 | 2020 | Yellow | 15 | 0.8705 |
| **Liang C. (2020)** | Liang C.; Chen T.; Dolman H.; Shi T.; Wei X.; Xu J.; Hagan D.F.T. | Drying And Wetting Trends And Vegetation Covariations In The Drylands Of China | Water (Switzerland), 12(4) | 2020 | Yellow | 6 | 0.3482 |
| **Tfwala C.M. (2021)** | Tfwala C.M.; Mengistu A.G.; Ukoh Haka I.B.; Van Rensburg L.D.; Du Preez C.C. | Seasonal Variations Of Transpiration Efficiency Coefficient Of Irrigated Wheat | Heliyon, 7(2) | 2021 | Yellow | 2 | 0.338 |
| **Suich H. (2017)** | Suich H.; Boardman J. | Wheat Growing And Changing Farming Systems In South African Dryland Margins: The Case Of The Sneeuberg, South Africa | Land Degradation And Development, 28(1), 57–64 | 2017 | Yellow | 2 | 0.0755 |
| **Brendel A.S. (2020)** | Brendel A.S.; Del Barrio R.A.; Mora F.; León E.A.O.; Flores J.R.; Campoy J.A. | Current Agro-Climatic Potential Of Patagonia Shaped By Thermal And Hydric Patterns | Theoretical And Applied Climatology, 142(3–4), 855–868 | 2020 | Yellow | 1 | 0.058 |
| **Hansen N.C. (2012)** | Hansen N.C.; Allen B.L.; Baumhardt R.L.; Lyon D.J. | Research Achievements And Adoption Of No-Till, Dryland Cropping In The Semi-Arid U.S. Great Plains | Field Crops Research, 132, 196–203 | 2012 | Purple | 122 | 1 |
| **Harper R.J. (2014)** | Harper R.J.; Sochacki S.J.; Smettem K.R.J.; Robinson N. | Managing Water In Agricultural Landscapes With Short-Rotation Biomass Plantations | Gcb Bioenergy, 6(5), 544–555 | 2014 | Purple | 17 | 1 |

**Table A2.** *Cont.*

| VOS Label | Authors | Title | Journal | Year | Cluster | Citations | Norm. Citations |
|---|---|---|---|---|---|---|---|
| **Wang E. (2009)** | Wang E.; Cresswell H.; Bryan B.; Glover M.; King D. | Modelling Farming Systems Performance At Catchment And Regional Scales To Support Natural Resource Management | Njas - Wageningen Journal Of Life Sciences, 57(1), 101–108 | 2009 | Purple | 22 | 0.5057 |
| **Hart B. (2020)** | Hart B.; Walker G.; Katupitiya A.; Doolan J. | Salinity Management In The Murray-Darling Basin, Australia | Water (Switzerland), 12(6) | 2020 | Purple | 8 | 0.4643 |
| **Acharya P. (2019)** | Acharya P.; Biradar C.; Louhaichi M.; Ghosh S.; Hassan S.; Moyo H.; Sarker A. | Finding A Suitable Niche For Cultivating Cactus Pear (Opuntia Ficus-Indica) As An Integrated Crop In Resilient Dryland Agroecosystems Of India | Sustainability (Switzerland), 11(21) | 2019 | Purple | 6 | 0.1978 |
| **Wang Y. (2018)** | Wang Y.; Gao F.; Yang J.; Zhao J.; Wang X.; Gao G.; Zhang R.; Jia Z. | Spatio-Temporal Variation In Dryland Wheat Yield In Northern Chinese Areas: Relationship With Precipitation, Temperature And Evapotranspiration | Sustainability (Switzerland), 10(12) | 2018 | Purple | 3 | 0.1543 |

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
