# Peer review of "Climate Change and Natural Resource Scarcity: A Literature Review on Dry Farming"

_land, doi:10.3390/land11122102_

Round 1

Reviewer 1 Report

MS: Climate change and natural resource scarcity: a literature review on dryfarming by Naomi di Santo and team.

Authors highlighted that agricultural sector is facing the challenge of climate change that drives to an increase in the difficulties to the activity and the economic sustainability of the primary sector, also affecting farmers’ revenues.

Following minor corrections require for further process-:

Abstract: Pl add quantify data

Keywords: Ok

Introduction: Add updated literature.

Conclusion: Add quantify data.

Overall, the manuscript has very information and may be consider after minor corrections. 

Author Response

Dear Reviewer, thank you for your valuable comments (in italics). We modified the original work following your suggestions. In the main document, the new version is in track changes.

 Abstract: Pl add quantify data

  1. We added information about the number of documents retrieved from the Scopus database.

 Introduction: Add updated literature.

  1. We wish to thank you for this observation. Actually, in the Introduction section the only “old” reference is from 1999 (and it is the oldest work cited in this section), while the others were mainly published after 2000, as it is the year that marked the increasing interest in this issue. The latest article is from the current year (2022).

Conclusion: Add quantify data.

  1. Thank you for your suggestion. We modified the conclusions accordingly.

Reviewer 2 Report

Comments

Results

Check Figure 2, why are there no papers published between 2005 and 2008?

Could authors include the list of all the journals as Appendix? See Table 1.

Conclusion

There should be a Section on “Policy Implications and Recommendations” after the Conclusions. Policy implications refer to what the results and findings and conclusions mean for policies and programmes of public sector and non-state actors. Policy implications are suggestive and indicative of broad and specific directions and instruments that can be used to address observed gaps, utilise observed opportunities and correct existing anomalies or inadequacies. Recommendations are what should be done, who should do it and in what conditions and for what. Recommendations are action-based and should be in the form of actionable measures. Recommendations should be actionable, sharp, unambiguous, reflective and specific.

Construct a set of policy implications around each conclusion or set of conclusions. With a good set of conclusions and policy implications, the stage is set for the identification of actionable recommendations.

There are no policy recommendations in the paper. Provide policy recommendations with actionable items for government, research institutions, development partners, private sector initiatives and civil society interventions. Be cautious not to make any conclusion that is not supported by evidence produced by the study. Also, avoid general statements about policy implications and provide links between conclusions and policy implications. No recommendations should be made that does not derive from the policy implications.

Authors should dedicate a section on discussing the limitations of the study and areas for further research. This section could come after Policy Implications and Recommendations.

References

Use the MDPI style of referencing.

Author Response

Dear Reviewer, thank you for your valuable comments (in italics). We modified the original work following your suggestions. In the main document, the new version is in track changes.

Results - Check Figure 2, why are there no papers published between 2005 and 2008?

  1. Thank you for your observation. As already argued in the paper, the relevance of the climate change and dry-land farming issues was highlighted starting from the introduction of Agenda 2030 (i.e., 2015). Moreover, following your comment, we analysed the trend in researches about climate change and dry-land farming separately. As far as climate change, we can say that before year 2006 there was a limited interest about this issue, that however improved afterwards when started an emerging and positive trend of research (Fig. 1A). With respect to dry-land farming, the papers on this topic show an increasing trend from 2015 (Fig. 1B) as now highlighted in the revised paper (in track changes).

Figure 1 - Trend in researches about "climate change" and "dry-farming" or "dry-land farming". Source: Scopus.

Could authors include the list of all the journals as Appendix? See Table 1.

  1. Thank you for this suggestion. We added in “Appendix” a table (table A) with the required information.

Conclusion - There should be a Section on “Policy Implications and Recommendations” after the Conclusions. Policy implications refer to what the results and findings and conclusions mean for policies and programmes of public sector and non-state actors. Policy implications are suggestive and indicative of broad and specific directions and instruments that can be used to address observed gaps, utilise observed opportunities and correct existing anomalies or inadequacies. Recommendations are what should be done, who should do it and in what conditions and for what. Recommendations are action-based and should be in the form of actionable measures. Recommendations should be actionable, sharp, unambiguous, reflective and specific. Construct a set of policy implications around each conclusion or set of conclusions. With a good set of conclusions and policy implications, the stage is set for the identification of actionable recommendations. There are no policy recommendations in the paper. Provide policy recommendations with actionable items for government, research institutions, development partners, private sector initiatives and civil society interventions. Be cautious not to make any conclusion that is not supported by evidence produced by the study. Also, avoid general statements about policy implications and provide links between conclusions and policy implications. No recommendations should be made that does not derive from the policy implications.

  1. Thank you for your valuable comment on Conclusions section that we amended accordingly. The original conclusion paragraph was split into separate paragraphs and the suggested sections were added.

Authors should dedicate a section on discussing the limitations of the study and areas for further research. This section could come after Policy Implications and Recommendations.

  1. Thank you for this suggestion. We added the suggested paragraph.

References - Use the MDPI style of referencing.

  1. Thank you for your observation. We reviewed the reference style.

Reviewer 3 Report

The article addresses a very important and timely issue. Agriculture is a major beneficiary of adverse climate change, even though its impact on the climate is disproportionately smaller than that of other economic sectors. The authors have reliably reviewed articles from the "Scopus" database on the issue under study, grouped them using an appropriate method. However, I have three main comments on the article presented:

1. The issue is covered very broadly, resulting in difficulties in drawing clear conclusions. The problems of drought and desertification are somewhat different in Africa, Asia, North America and Europe. In addition, the wealth of each country varies, and the possibility of influence through agricultural and environmental policy instruments also varies.

2. I feel that the main goal of the work, which was to identify advice for policy makers, including economic and financial advice, on the main needs arising from climate change and already identified by researchers in other disciplines, has not been achieved. Too much spatial variation results in a lack of specific guidance.

3. The conclusions presented are very general. Not much can be concluded from them.

I believe that the article could be printed with a clarification of the conclusions of the research and an indication of the need for more detailed research in groups of more homogeneous regions in terms of climate conditions and economic development.

Author Response

Dear Reviewer, thank you for your valuable comments (in italics). We modified the original work following your suggestions. In the main document, the new version is in track changes.

  1. The issue is covered very broadly, resulting in difficulties in drawing clear conclusions. The problems of drought and desertification are somewhat different in Africa, Asia, North America and Europe. In addition, the wealth of each country varies, and the possibility of influence through agricultural and environmental policy instruments also varies.
  2. I feel that the main goal of the work, which was to identify advice for policy makers, including economic and financial advice, on the main needs arising from climate change and already identified by researchers in other disciplines, has not been achieved. Too much spatial variation results in a lack of specific guidance.
  3. The conclusions presented are very general. Not much can be concluded from them. I believe that the article could be printed with a clarification of the conclusions of the research and an indication of the need for more detailed research in groups of more homogeneous regions in terms of climate conditions and economic development.
  4. Thank you for your very valuable observation. We have modified, where possible the original paper (please see in track changes). However, the original aim of the study was to carry out an objective and replicable bibliometric analysis: this required the consideration of a huge number of articles, thus it would not have been possible splitting the analysis to deepen the study focusing in different Countries. In our view, the aim of focusing on a specific Country or homogeneous group of Countries, to give very specific suggestions to policy makers, would have been better addressed with a systematic or a scoping review. Conversely, the main weaknesses of these two latter methodologies are represented by risk of bias and broader and less objective results.

Regarding the conclusions, we modified the section also adding some paragraphs on policy implications, limitations of the study and possible fields or future research. As you highlighted, the issue could show different aspects in different countries, so we suggested to carry out future researches investigating and addressing the topic in specific territories, so to be able to formulate appropriate policy suggestions for the specific territorial framework.

Round 2

Reviewer 2 Report

The manuscript should be subjected to English language editing

Author Response

We have made the English revision as require..
